# A Brain-Inspired Regularizer for Adversarial Robustness

## Abstract

Convolutional Neural Networks (CNNs) excel in many visual tasks, but they tend to be sensitive to slight input perturbations that are imperceptible to the human eye, often resulting in task failures. Recent studies indicate that training CNNs with regularizers that promote brain-like representations, using neural recordings, can improve model robustness. However, the requirement to use neural data severely restricts the utility of these methods. Is it possible to develop regularizers that mimic the computational function of neural regularizers without the need for neural recordings, thereby expanding the usability and effectiveness of these techniques? In this work, we inspect a neural regularizer introduced in Li et al. (2019) to extract its underlying strength. The regularizer uses neural representational similarities, which we find also correlate with pixel similarities. Motivated by this finding, we introduce a new regularizer that retains the essence of the original but is computed using image pixel similarities, eliminating the need for neural recordings. We show that our regularization method 1) significantly increases model robustness to a range of black box attacks on various datasets and 2) is computationally inexpensive and relies only on original datasets. Our work explores how biologically motivated loss functions can be used to drive the performance of artificial neural networks.

## 1 Introduction

Convolutional Neural Networks (CNNs) have achieved high performance on a variety of visual tasks such as image classification and segmentation. Despite their remarkable success, these networks are notably brittle; even a small change in the input can significantly alter the network's output Biggio et al. (2013); Szegedy et al. (2013). Szegedy et al. (2013) found that small perturbations, imperceptible to the human eye, can lead CNNs to misclassify images. These adversarial images pose a significant threat to computer vision models.

Improving the robustness of CNNs against adversarial inputs is a major focus in machine learning. Various methods have been proposed, each with different levels of success and computational demands Li et al. (2022). Some researchers have drawn inspiration from the mammalian brain, finding that deep neural networks trained to mimic brain-like representations are more resistant to adversarial attacks (Li et al., 2019; Safarani et al., 2021; Li et al., 2023). In particular, Li et al. (2019) demonstrated that incorporating a regularizer into the loss function, which aligns the CNN's representational similarities (Kriegeskorte et al., 2008) with those of the mouse primary visual cortex (V1), significantly enhances the network's robustness to Gaussian noise and adversarial attacks. Using a loss term to steer models towards brain-like representations is referred to as neural regularization. However, a significant drawback of these methods is the reliance on neural recordings, which are often difficult to obtain and limit the methods' applicability.

In this work, focusing on CNNs used for image classification tasks, we take a deeper look at the neural regularizer introduced in Li et al. (2019). We ask whether the underlying working principles of this biologically-inspired regularizer can be extracted and utilized to enhance the robustness of deep neural networks without relying on large-scale neural recordings. In particular, the regularizer introduced in Li et al. (2019) steers a CNN's representations to match the representational similarities of a predictive model of brain recordings. Taking this as a starting point, our contributions are the following:

- We observe that the representational similarities produced by the predictive model correlate highly with pixel-based similarities. Motivated by this, we propose a simple and interpretable similarity measure for regularization derived from the regularization image dataset, without using neural data.

- We evaluate the robustness of regularized models on black box attacks, where the attacker only has query access to the model as opposed to white box attacks where gradients and parameters are accessible to the attacker. We show that our regularizer drives the network to be more robust to a wide range of black box attacks.

- We demonstrate the flexibility of our method by using different datasets for regularization, including the classification dataset itself. We show that our method works on both grayscale and color datasets.

- We assess the robustness of the regularized models to common corruptions using the CIFAR-10-C dataset Hendrycks & Dietterich (2019a).

- Our regularization method is computationally efficient, relying on original image datasets without the need for data distortions or augmentations during training. We show that it requires a relatively small regularization batch size and a small number of regularization images.

- We show that our regularization method primarily protects against high-frequency perturbations by analyzing the Fourier transformation of minimal perturbations needed to mislead models, as obtained from decision-based Boundary Attacks Brendel et al. (2017).

Our work demonstrates that a brain-inspired regularizer can enhance model robustness without large-scale neural recordings. This contributes to the broader use of biologically-inspired loss functions to improve artificial neural networks' performance. The end product is a simple, computationally efficient regularizer that performs well across a wide range of scenarios.

## 2 RELATED WORKS

**Adversarial attacks.** Identifying adversarial examples that can mislead a model is a dynamic field of research, with an increasing number of attacks being introduced Szegedy et al. (2013); Hinton et al. (2015); Moosavi-Dezfooli et al. (2016); Brendel et al. (2017); Madry et al. (2017). In this study, we concentrate on black box attacks, which do not have access to detailed model information, as these are more reflective of real-world scenarios. We evaluate our models against four types of attacks: random noise, common corruptions, transfer-based attacks, and decision-based attacks.

Random noise attacks involve applying noise sampled from known distributions (e.g., Gaussian, Uniform, and Salt and Pepper) to an input - see Appendix A.1. Common corruptions correspond to distortions that can be found in real life computer vision applications (eg : motion blur) Hendrycks & Dietterich (2019a). A transfer-based attack involves finding adversarial perturbations for a substitute model (an unregularized model in our case) and applying them to a target model. Evaluating robustness on transfer-based attacks is crucial because adversarial examples crafted for one model can also mislead another distinct model Papernot et al. (2016). We find these perturbations by applying the Fast Gradient Sign Method (FGSM) Goodfellow et al. (2014) to the substitute model. Such adversarial samples are computed as :

$$x_{adv} = x + \epsilon \times \text{sign}(\nabla_x L(\theta, x, y)) \tag{1}$$

where $x_{adv}$ is the adversarial example, $x$ denotes the original input image, $y$ denotes the original input label, $\theta$ denotes the model parameters and $L$ is the loss. A decision based attack is an attack which solely depends on the final model's decision. Evaluating robustness on them is key as they are applicable to real world black box models. Precisely, we evaluate robustness on a Boundary Attack introduced by Brendel et al. (2017). This attack starts from a large adversarial perturbation, and seeks to reduce the perturbation while remaining adversarial.

**Adversarial training as a defense to adversarial attacks.** As adversarial attacks have advanced, corresponding defenses have also been developed to secure against them Goodfellow et al. (2014); Bhagoji et al. (2018); Diffenderfer et al. (2021); Kireev et al. (2022). A common defense strategy involves augmenting each batch of training data with adversarial examples, a technique known as adversarial training Goodfellow et al. (2014); Madry et al. (2017). A widely-used method for

generating these adversarial examples is the Projected Gradient Descent (PGD) attack Madry et al. (2017), which is a multi-step variant of the Fast Gradient Sign Method (FGSM) attack. PGD is popular due to its effectiveness in creating challenging adversarial examples, thereby enhancing the model's resilience against attacks.

**Neural regularization.** Recent work showed that jointly training a deep network to perform image classification while steering it towards having brain-like representations can improve the model's robustness to adversarial attacks Li et al. (2019); Safarani et al. (2021). This is achieved by introducing a penalty term in the loss function acting directly on image representations Li et al. (2019) or activations Safarani et al. (2021) at different network depths. Such a process is referred to as neural regularization. For instance, Li et al. (2019) used a neural regularizer to drive a CNN to aligh its representational similarities (Kriegeskorte et al., 2008) with those of mouse primary visual cortex (V1). Later on, Safarani et al. (2021) used a neural regularizer to drive a CNN towards predicting neural activity in macaque primary visual cortex (V1) in response to the same natural stimuli. The key bottleneck of such defenses is their reliance on the measurement of large scale neural recordings.

## 3 A NEURAL REPRESENTATIONAL SIMILARITY REGULARIZER

To increase the robustness of artificial neural networks to adversarial attacks, one research direction focuses on extracting and applying computational concepts from the mammalian brain. In particular, Li et al. (2019) showed that adding a neural regularizer term to the training loss enhances the adversarial robustness of CNNs on image classification tasks. The regularization term is denoted by $L_{\text{sim}}$ as it depends on similarities between neural responses.
The loss function $L$ is written as:

$$L = L_{\text{task}} + \alpha L_{\text{sim}} \tag{2}$$

where $L_{\text{sim}}$ given image pairs $(i, j)$ is defined as,

$$L_{\text{sim}} = \sum_{i \neq j} \left( \text{arctanh}(S_{ij}^{\text{CNN}}) - \text{arctanh}(S_{ij}^{\text{target}}) \right)^2, \tag{3}$$

and $\alpha$ is a parameter that sets the overall regularization strength. $S_{ij}^{\text{target}}$ in eq. equation 3 is the target's pairwise cosine similarity between the representations of images $i$ and $j$. $S_{ij}^{\text{CNN}}$ measures the similarity between the representations of images $i$ and $j$ in a CNN. We compute it following the approach of Li et al. (2019). We combine feature similarities from a selection of $K$ equally spaced convolutional layers and average the results through a *trainable* weights $\gamma_l$, where $l$ is the layer number. The latter are the output of a softmax function meaning $\sum_l \gamma_l = 1$ and $\gamma_l \geq 0$. Overall,

$$S_{ij}^{\text{CNN}} = \sum_l \gamma_l S_{ij}^{\text{CNN}-l}, \tag{4}$$

where $S_{ij}^{\text{CNN}-l}$ is the mean-substracted cosine feature similarity between images $i$ and $j$ at layer $l$. Having $\gamma_l$ be trainable enables the model to choose which layer(s) to regularize to match the similarity target.

In their setup, Li et al. (2019) used a ResNet (He et al., 2016) to classify grayscale CIFAR-10 and CIFAR-100 datasets. To compute $S_{ij}^{\text{target}}$, neural responses were collected from mouse primary visual cortex (V1), while the mouse was looking at grayscale images from ImageNet dataset. However, in practice, due to noise in neural recording, $S_{ij}^{\text{target}}$ was not computed from the neural recordings directly. Instead, it was computed from a predictive model (Sinz et al., 2018; Walke et al., 2018) trained to predict the neural responses from images. The predictive model consisted of a 3-layered CNN with skip connections Sinz et al. (2018); Walke et al. (2018); Li et al. (2019), it accounts for behavioural data such as pupil position, size and running speed on treadmill (Appendix A.1).

Training is done by first processing a batch of images from the classification task dataset to calculate the classification loss $L_{\text{task}}$, and then processing a batch of image pairs from the regularization dataset to compute the similarity loss $L_{\text{sim}}$. We then compute the full loss $L$ which we use for backpropagation.

We inspect the similarity loss $L_{\text{sim}}$ (eq.equation 3) introduced in Li et al. (2019) to extract its underlying strength. Our goal is to formulate a method that can bypass the use of neural recordings which can be costly.

Since the primary visual cortex (V1), where the neural recordings come from, is the first visual processing area in the cortex, we inspect the correlation between the neural representational similarity and image pixel similarities (computed as described in Appendix A.1). We observe that there is a high correlation between the two as shown in Fig. 2, left panel. Thus, we investigate the effect of using pixel similarities as target similarities in $L_{\text{sim}}$ instead of similarities obtained from neural recordings. To compare both approaches, we replicate the experimental setup in Li et al. (2019). We train a ResNet to classify grayscale CIFAR-10 and CIFAR-100, and use grayscale images from ImageNet data, as the regularization dataset (same datasets used in (Li et al., 2019)). However, we differ from Li et al. (2019) in our choice of $S_{ij}^{\text{target}}$: we set $S_{ij}^{\text{target}}$ in eq. equation 3 to $S_{ij}^{\text{pixel}}$, where $S_{ij}^{\text{pixel}}$ is computed as the pixel cosine similarity between images which are flattened, mean subtracted and normalized. After training, we observe that the regularized model exhibits some enhancement in robustness, however this enhancement is not consistent across different image perturbations and adversarial attacks. For example, we see a modest enhancement in robustness to Gaussian noise (Fig. 1a), and decision-based Boundary Attack (Fig. 1c), but this was not the case for transferred FGSM attack (Fig. 1b). For Uniform noise and Salt and Pepper noise, we see an enhancement in robustness at large noise levels (see Fig. 12 in A.2). In Fig. 1 we also show the performance of the model of Li et al. (2019), which is regularized using neural representational similarities, for direct comparison.

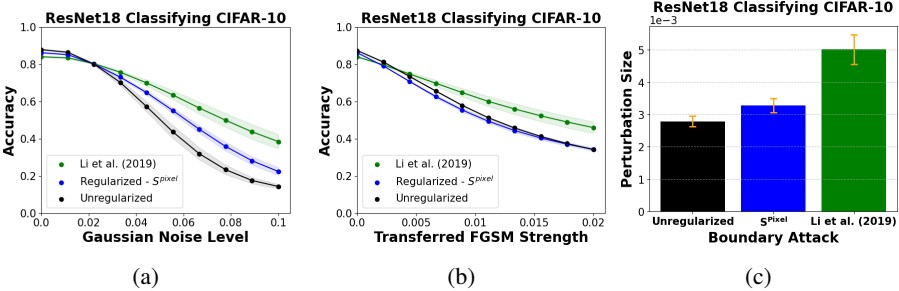

(a)           (b)           (c)

Figure 1: ResNet18 classifying grayscale CIFAR-10 is regularized using ImageNet ($S^{\text{pixel}}$) or a neural predictive model trained on ImageNet as in Li et al. (2019). (a) Robustness to Gaussian Noise, (b) Transferred FGSM Goodfellow et al. (2014) perturbations and (c) to a decision-based Boundary Attack Brendel et al. (2017) as shown by the difference in retrieved perturbation sizes retrieved. Neural and pixel regularized models use $\alpha = 10$. Error shades/bars represent the SEM across 7 seeds per model - except for neural models where we had access to 5 models. (Details on the experimental setup can be found in Appendix A.1)

If we go back and visually examine the last two panels in Fig.2, we observe that the image pixel and neural representational similarity matrices have similar patterns, however the pattern is more enhanced in the neural representational similarity matrix. Thus, at the first stage of cortical visual processing, the brain seems to roughly preserve an underlying structure of the image pixel similarities but amplify it. Based on this observation, we define a new target similarity $S^{Th}$, such that

$$S_{ij}^{Th} = \begin{cases} 1, & \text{if } S_{ij}^{\text{pixel}} > Th, \\ -1, & \text{if } S_{ij}^{\text{pixel}} < -Th, \\ 0, & \text{if } |S_{ij}^{\text{pixel}}| \leq Th \end{cases} \quad (5)$$

where, $Th \in (0, 1)$ is a tunable thresholding hyperparameter. That is, we set $S_{ij}^{\text{target}}$ to be $S_{ij}^{Th}$. We note that in practice we don't use exactly 1 and $-1$ in eq. equation 5 by a very small number $\epsilon$ because of the arctanh function in eq.equation 3. Finally, we note that even though we use $L_{\text{sim}}$ as in equation 3, the application of the arctanh function is not necessary in this case. The intuition behind this regularization term is as follow: it is constraining the lower layers in the network (as we show in Appendix A.6) to have identical representations for image pairs that are close in pixel space, measured by the cosine similarity; hence viewing those images pairs as adversarial versions of each other. At the same time, it is pushing images farther away in pixel space to have orthogonal representations. In A.7, we show the contribution of each term of $S^{Th}$ on robustness. In section 4.2 we propose how to select the hyperparameter $Th$ and the regularization hyperparameter $\alpha$.

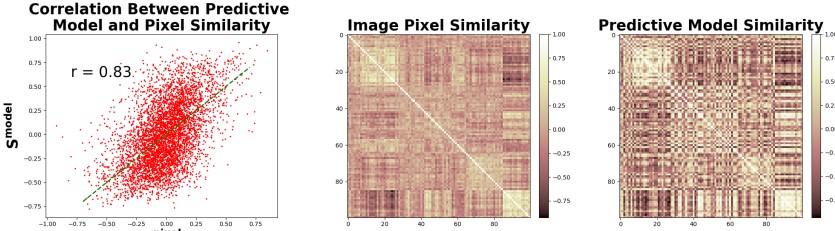

Figure 2: Correlation between representational similarities computed from the neural predictive model used in Li et al. (2019) (see section 3 and Appendix A.1) and from image pixels computed as the cosine similarity between images which are flattened, mean subtracted and normalized. We trained more than 3 models on 6 distinct scans to predict neural responses and averaged their resulting representational similarity. We observe that the neural representational similarity correlates with the image pixel similarity.

Key advantages for this regularization method are 1) it does not require access to large scale neural recordings, 2) it relies on the original datasets, and does not require the introduction of data distortions or augmentations during training, 3) it is computationally inexpensive, as we show later in Section 4.6

## 4 EXPERIMENTS

We train ResNets He et al. (2016) on image classification tasks using $L = L_{\text{task}} + \alpha\, L_{\text{sim}}$ (eq. equation 2, and setting $S^{\text{target}} = S^{Th}$ in the regularization in the loss $L_{\text{sim}}$ (eq. equation 3). The $(\alpha, Th)$ pairs used for each classification-regularization dataset pairs are reported in Appendix A.5. We also report in Appendix A.6 the value of $\gamma_l$ (as defined in Section 3) for each dataset combination. After training, we evaluate the regularized model robustness to a set of black box adversarial attacks (Section 2). Even though we report below results for ResNet18, we show in A.4 and A.8 results for ResNet34. To allow direct comparison with Li et al. (2019), we mainly show results using grayscale CIFAR-10 as classification dataset. However, to demonstrate the success of our method, we also show results using colored CIFAR-10. Furthermore, in the appendix we show results using other classification datasets like grayscale CIFAR-100 (A.4), colored CIFAR-100 (A.8), MNIST (A.4) and FashionMNIST (A.4). The details of our experimental setup and implementation can be found in Appendix A.1.

### 4.1 ROBUSTNESS TO ADVERSARIAL ATTACKS

We first test the robustness of regularized models using grayscale CIFAR-10 and grayscale ImageNet as classification and regularization datasets respectively, as used in Li et al. (2019). We first show robustness to Gaussian noise perturbations. We find that regularized models exhibit a substantial increase in robustness when compared to unregularized models as seen in Fig.3, left panel. They also show a similar performance to neural regularized models Li et al. (2019) (Fig.3, left panel). The robustness of models regularized using $S^{Th}$, to Uniform and Salt and Pepper perturbations can be found in appendix A.3.

We then test robustness to stronger black box attacks, particularly, to transferred FGSM (Goodfellow et al., 2014) perturbations from an unregularized model, and decision-based Boundary Attack Brendel et al. (2017) (Section 2). We observe an increase in robustness to both attacks (Fig.3, center and right panels). Note that for decision-based Boundary attack, the larger the perturbation size between the adversarial input and the original image, the better in terms of robustness. Again, we observe that models regularized using $S^{th}$ perform similar to those regularized using neural data Li et al. (2019) (Fig. 3, center and right panels).

The experiments above demonstrate that we can obtain similar robustness to neural regularized models Li et al. (2019) by simply regularizing using $S^{Th}$, which does not require neural data, and relies only on the original unaugmented regularization dataset.

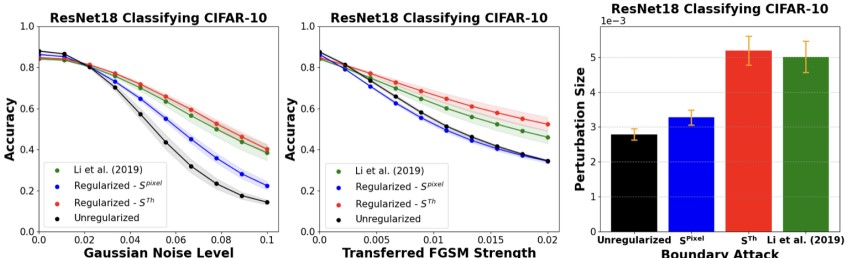

Figure 3: Robustness to Gaussian noise (left), transferred FGSM Goodfellow et al. (2014) (center), and decision-based Boundary Attack Brendel et al. (2017) (right). A ResNet18 is trained to classify grayscale CIFAR-10 and is regularized on grayscale images from ImageNet dataset. Results for different regularization targets are shown : $S^{pixel}$, $S^{Th}$ and neural based targets as in Li et al. (2019). For the decision-based Boundary Attack, we compute the median squared $L_2$ perturbation size per pixel, averaged across 1000 images, and 5 repeats. Error shades represent the SEM across seven seeds per model.

## 4.2 HYPERPARAMETER SELECTION AND CONSISTENT BEHAVIOR ACROSS ATTACKS

An important question is how to select an $\alpha, Th$ hyperparameter pair ? We propose a criteria to select those hyperparameters, as follows. A suitable pair should (1) be such that the resulting model has an 'acceptable' accuracy on the distortion-free dataset, and (2) showcases an increase in robustness to adversarial attacks. To properly define what we mean by this, we introduce the following quantities $R_0, R_N, U_0$ and $U_D$. Where, $R_0$ is the regularized model's accuracy on distortion-free images and $R_D$ its accuracy at high distortion level. $U_0$ and $U_D$ are their equivalent for the unregularized model. The ratios $\frac{R_0}{U_0}$ and $\frac{R_D}{U_D}$ reflect how our regularization affects the model's accuracy at zero and high distortion levels. To meet condition (1), we require that $\frac{R_0}{U_0} \geq A_0$, where $A_0$ is user defined. We select $A_0 = 0.9$. Condition (2) is simply met by requiring that $\frac{R_D}{U_D} > 1$.

We can visualize the performance of a model by plotting $\frac{R_0}{U_0}$ vs $\frac{R_D}{U_D}$ for each $\alpha, Th$ pair. This allows the user to select the hyperparameter pair based on the selection criterion that they choose. In Fig.4 we show the above plot for different adversarial attacks (the gray shaded planes).
As seen, the regularization method produces a consistent behavior across all the adversarial attacks that we use. This allows the user to use the simplest attack, like adding Gaussian noise to the images, to select the $\alpha, Th$ pair. In Fig.4, the blue shaded area represents the region where conditions (1) and (2) are met for each attack.

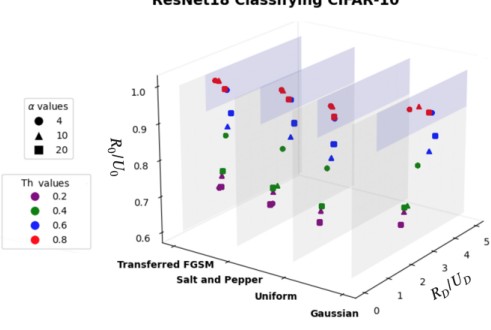

Figure 4: Behavior across multiple black box attacks and hyperparameters $(\alpha, Th)$ choices. Models are trained to classify grayscale CIFAR-10 and regularized on grayscale images from ImageNet dataset. Different planes show results for different black box attacks. In each plane, the region shaded in blue represents the region of 'acceptable' models, which we've taken here to be $R_D/U_D \geq 1$ and $R_0/U_0 \geq 0.9$ – a criteria that can be adjusted as needed. $U_D, R_D$ are computed at $\epsilon = 0.1$ for random attacks and $\epsilon = 0.02$ for the transferred FGSM Goodfellow et al. (2014) attack. Mean metrics across 7 seeds per model are displayed.

## 4.3 ROBUSTNESS ACROSS DATASETS COMBINATIONS

Our method is flexible to the choice of the regularization dataset. We find that regularizing on different datasets leads to an increase in model robustness, but, there is a quantitative difference in the robustness level achieved by regularizing on different datasets. Fig. 5 shows the performance of a ResNet18 trained to classify grayscale CIFAR-10 regularized on grayscale images from three datasets separately (CIFAR-10, CIFAR-100 or ImageNet) for three attacks (Gaussian noise, transferred FGSM, and decision-based Boundary Attack) compared to an unregularized model. In Appendix A.4 we show results for different classification-regularization datasets combinations.

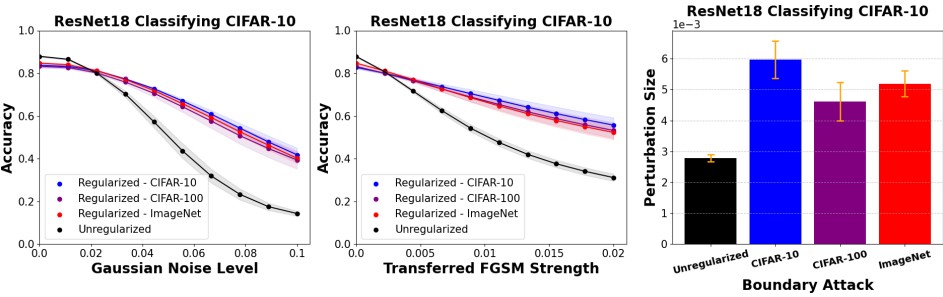

Figure 5: Robustness of a ResNet18 trained to classify grayscale CIFAR-10 regularized on grayscale images from different datasets : grayscale CIFAR-10 (blue), CIFAR-100 (purple) or ImageNet (red). For the decision-based Boundary Attack, we compute the median $L_2$ perturbation size, averaged across 1000 images, and 5 repeats. Error shades/bars represent the SEM across seven seeds per model. The same $(\alpha, Th)$ values were used in training all models i.e for all regularization datasets (see Appendix A.5).

## 4.4 ROBUSTNESS TO COMMON CORRUPTIONS

Regularized models are also more robust than their unregularized counterpart on common corruptions. We evaluate regularized models on grayscale CIFAR-10-C dataset Hendrycks & Dietterich (2019a) which consists of grayscaled CIFAR-10 images with common corruptions that can be found on everyday computer vision application. Evaluating on common corruptions at different severity levels is critical as they simulate real world conditions. Fig. 6 shows the performance of a ResNet18 trained to classify CIFAR-10 regularized with grayscale images from ImageNet dataset vs unregularized model. Fig. 6 (left) shows the performance averaged over all 15 common corruptions at different severity levels, Fig. 6 (right) shows the robustness of unregularized and regularized models for the 15 individual corruptions present in CIFAR-10-C Hendrycks & Dietterich (2019a), at severity level 4.

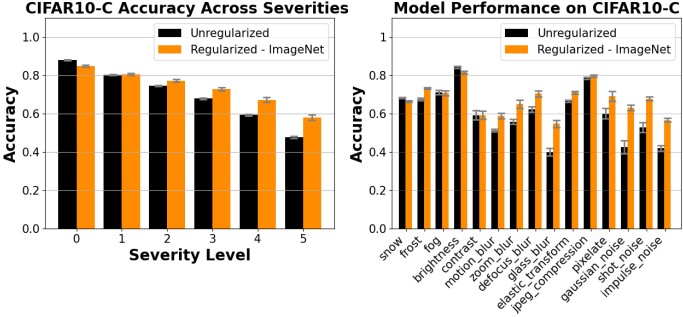

Figure 6: Robustness to grayscale CIFAR-10-C Common Corruptions Hendrycks & Dietterich (2019a). (left) We compute the regularized model accuracy on grayscale CIFAR-10-C for different severity levels, averaging across all 15 common corruptions present in CIFAR-10-C. (right) We show the robustness of unregularized and regularized models on 15 individual corruptions at severity 4. Error bars correspond to the SEM across seven seeds per model. Results are for a ResNet18 trained to classify grayscale CIFAR-10 regularized wih grayscale images from ImageNet dataset.

### 4.5 FREQUENCY DECOMPOSITION OF ADVERSARIAL PERTURBATIONS AND COMMON CORRUPTIONS

To understand the strengths and weaknesses of our regularization method, we investigate the frequency components present in the minimal perturbation required to flip the decision of unregularized and regularized models which we compute *via* a decision-based Boundary Attack Brendel et al. (2017). We observe that models regularized using pixel-based similarities ($S^{Th}$) rely more on low frequency information than their unregularized counterparts (Fig. 7 center and right panels). We further evaluate our regularized model performance on grayscale CIFAR-10-C Hendrycks & Dietterich (2019a) following the approach described in Li et al. (2023), where we categorize the 15 corruptions in CIFAR-10-C into Low, Medium, and High frequency based on their spectra (see Fig. 7 left panel and Appendix Fig. 25). Results are shown for ResNet18 trained to classify CIFAR-10 and regularized using images from CIFAR-10, CIFAR-100 or imageNet datasets. Our results show that regularized models outperform unregularized ones, especially on high-frequency corruptions, confirming our findings. Such a reliance on low-frequency information has also been observed in models subjected to neural regularization as explained in Li et al. (2023).

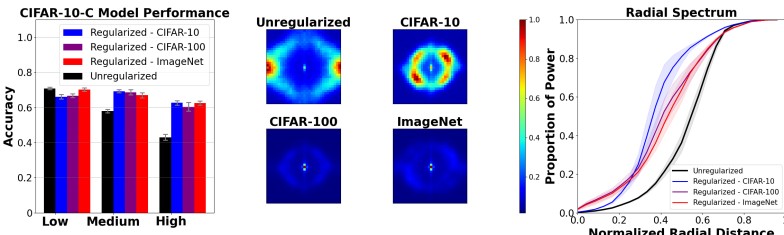

Figure 7: Frequency perspective on robustness. The results are for ResNet18 trained to classify grayscale CIFAR-10 and regularized on grayscale images from different datasets: CIFAR-10, CIFAR-100 or ImageNet. (left) Robustness of regularized ResNet18 models evaluated on grayscale CIFAR-10-C at severity 4, categorized by the frequency range of each corruption. (center) Fourier power spectrum for the mean minimal corruption required to flip a model's decision. (right) Radial Spectrum of minimal pertubation required to mislead models, as provided by a decision-based Boundary Attack Brendel et al. (2017) - using 10k steps. The error bars (left panel) and shaded areas (right panel) represent the SEM across seven and four seeds per model respectively.

### 4.6 COMPUTATIONAL ADVANTAGES

In addition to being a simple method to apply, our regularization method is computationally inexpensive. First, in regard to training time, for $k$ image pairs per regularize batch, the additional time taken per batch to train the model corresponds to $2 \times k$ additional forward passes. We see in Fig. 8 that the method is successful for regularization batch size values: 4, 8, 16, 32. Choosing a smaller batch size can help in cutting the extra training time needed for successful regularization.

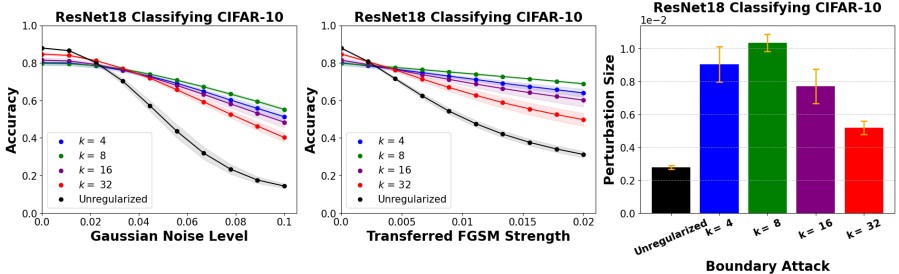

Figure 8: Robustness of a ResNet18 trained to classify grayscale CIFAR-10 and regularized on grayscale images from ImageNet dataset is shown for different regularization batch sizes, $k \in \{4, 8, 16, 32\}$. For the decision-based Boundary Attack, we compute the median $L_2$ perturbation size, averaged across 1000 images, and 5 repeats. Error shades/bars represent the SEM across seven seeds per model.

Second, although the number of target similarities is $\binom{N}{2}$ for $N$ selected regularization images, we find that we do not need many images for regularization. In our experiments we used $N = 5000$, leading to approx $12 \times 10^6$ pairs (Appendix A.1), however, we show in Fig. 9 that we do not need that much images; as can be seen using only $N \in \{100, 1000\}$ images provides robustness increase to black box attacks. Last, our method relies on the original image datasets, and does not require the introduction of different data distortions or augmentations during training.

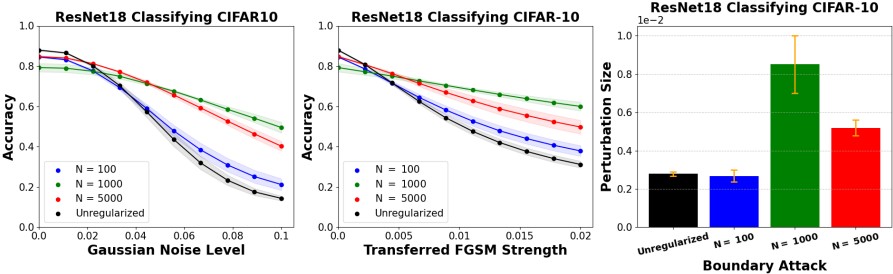

Figure 9: Robustness of a ResNet18 trained to classify grayscale CIFAR-10 and regularized on grayscale images from ImageNet dataset is shown for different number of regularization images. For the decision-based Boundary Attack, we compute the median $L_2$ perturbation size, averaged across 1000 images, and 5 repeats. Error shades/bars represent the SEM across seven seeds per model.

### 4.7 RESULTS USING COLOR DATASETS

Our previous results were obtained using grayscale datasets, which as we previously mentioned, were chosen to allow direct comparison with Li et al. (2019), and for consistency. Here, we show that our method is also successful when using color datasets, which are more utilized in practice.

In Fig. 10 we show results using color CIFAR-10 as classification dataset, and color CIFAR-10, CIFAR-100 or ImageNet as regularization datasets. As seen, there is an increase in the model's robustness for all regularization datasets. Similar results are observed when using color CIFAR-100 as classification dataset (see Appendix A.8).

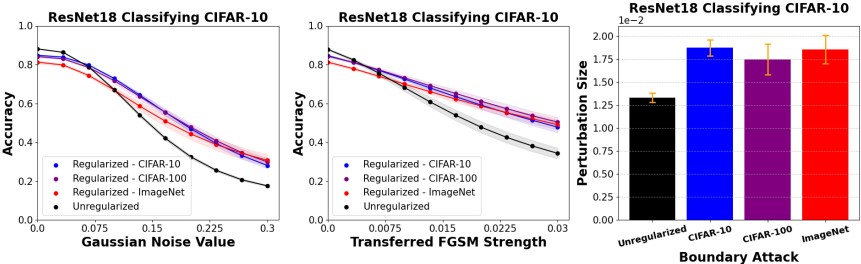

Figure 10: Robustness of a ResNet18 trained to classify colored CIFAR-10 regularized on colored images from different datasets : CIFAR-10 (blue), CIFAR-100 (purple) or ImageNet (red). For the decision-based Boundary Attack, we compute the median $L_2$ perturbation size, averaged across 1000 images, and 5 repeats. Error shades/bars represent the SEM across seven seeds per model.

## 5 CONCLUSION AND DISCUSSION

Extracting the working principles of the brain to advance AI is a long-term goal of neuroscience. To further this goal, we examined a brain-inspired method for adversarial robustness proposed by Li et al. (2019). This method uses neural recordings from the brain to align learned representations in an artificial neural network with brain representations through a regularization term added to the training loss. We extracted the core working principle behind this regularizer and proposed a simple, pixel-based regularization scheme that achieves similar performance, and gave an intuitive interpretation of our method. These findings contribute to the broader objective of leveraging

brain-inspired principles to advance AI.

We showed that our proposed method increases the robustness of CNNs to a spectrum of black box attacks (Section 2). We proposed a method to select the regularization hyperparameters $(\alpha, Th)$. We also showed that the choice of an $(\alpha, Th)$ pair value for regularization, affects the robustness level in a consistent way across different attacks. We demonstrated the effectiveness and scalability of our method, by showing its success in increasing model robustness using different combinations of classification and regularization datasets, including classification datasets CIFAR-10 and CIFAR-100 (Appendix A.1). We evaluated the performance of regularized models on common corruptions using grayscale CIFAR-10-C. We performed a Fourier analysis on minimal adversarial perturbations obtained from a decision-based Boundary Attack on our regularized model, and found that the perturbations from the regularized model contained higher low-frequency components relative to the unregularized model. We also showed that our method is more effective against common corruptions that are categorized as high frequency corruptions based on the average frequency estimated from the Fourier spectrum of the perturbations induced by these corruptions Li et al. (2023). These findings are in line to those in Li et al. (2023), who examined the same perturbations for a model regularized using neural data Li et al. (2019). Even though we mostly presented results using grayscale datasets to allow direct comparison with the method in Li et al. (2019), we demonstrated that our method is also successful when using color datasets, where we showed results for color CIFAR-10 and color CIFAR-100. We also investigated the contribution of different parts of $S^{Th}$ (Appendix A.7).

Even though we use a biologically inspired loss term that originally utilized large scale neural recordings to enhance the robustness of machine learning models, we have shown that this loss term can be implemented in a successful way that bypasses the use of neural data. Furthermore, our regularization method, although effective, is very simple. It relies on the original image datasets, and does not require the introduction of any additional data distortions or augmentations during training. It is flexible in regard to choosing the regularization dataset. It is computationally inexpensive, it requires a relatively small batch size for regularization. It also requires a small number of images to regularize on, or more precisely to construct the targets $S^{Th}$ (eq. equation 5). Our work is an encouraging step towards dissecting the workings of neural regularizers, to come up with methods that can both, enhance the performance of machine learning models, and be implemented by a broader machine learning community. Finally, we point out that one limitation of our method, is its inability to increase model robustness to some common corruptions as can be seen in Fig. 6 (right) and Fig .7 (left). Also, it does not achieve the level of robustness attained using state of the art defenses against adversarial attacks Croce et al. (2020). Although we stress that our aim is not to come up with the best adversarial defence, but rather to show that our method which is based on the neural regularizer in Li et al. (2019) can be equally effective without the need to use expensive neural data.

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

# A APPENDIX / SUPPLEMENTAL MATERIAL

## A.1 EXPERIMENTAL SETUP

**Training - Neural Predictive Model** - We train neural predictive models Sinz et al. (2018); Walke et al. (2018) to predict neural responses using all scans (measurements of neurons) with the same training configurations and neural data available in the codebase left by Li et al. (2019).

**Training - Image classification** - All models were trained by stochastic gradient descent on a NVIDIA A100-SXM4-40GB GPU. Models classifying grayscale CIFAR-10 were trained during 40 epochs. Training and regularizing a ResNet18 CIFAR-10 took in average 34 min to run. We used a batch size of 64 for the classification pathway and a batch of 16 image pairs for the regularization pathway. The number of regularization images used by default is $5,000$. Target similarities $S_{ij}^{\text{pixel}}$ are computed as follows. We compute the cosine similarity between images which are flattened, mean subtracted and normalized. We use the same learning schedule as in Li et al. (2019). Models were trained using Pytorch (Paszke et al., 2017).

The classification datasets used are : MNIST Lecun et al. (1998), FashionMNIST Xiao et al. (2017), grayscale CIFAR-10 Krizhevsky et al. (2009), grayscale CIFAR-100. The regularization datasets used are : MNIST, FashionMNIST, grayscale CIFAR-10, grayscale CIFAR-100 and grayscale ImageNet Deng et al. (2009). In Appendix A.5, we report the $(\alpha, Th)$ used for each dataset combinations. The codebase used for training is based on the codebase used in Li et al. (2019). All training codes are supplemented with this submission.

**Adversarial attacks** (see Section 2) - All perturbations are reported for image pixels in the range $[0, 1]$. We evaluate model robustness to random noise and transferred FGSM Goodfellow et al. (2014) perturbations by measuring the accuracy of evaluated models for distinct perturbation strengths $\epsilon$. We empirically find the range of perturbation strengths used to evaluate models, such that the unregularized model performs bad at the highest $\epsilon$ used for that particular model-dataset-attack combination. The random noise perturbations are showcased in Fig 11.

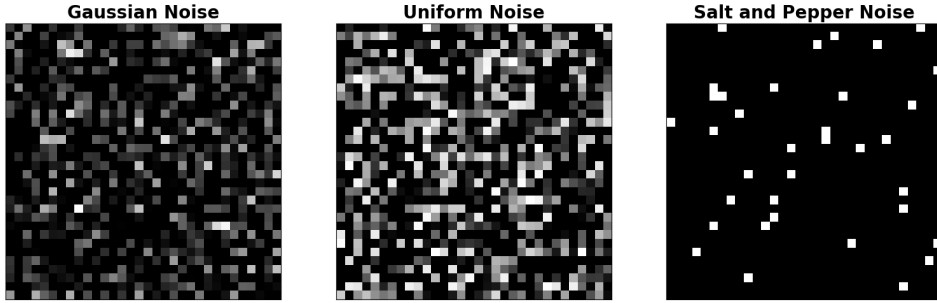

Figure 11: Visualization of (left) Gaussian noise, (center) Uniform noise and (right) Salt and Pepper noise with $\epsilon = 0.06$.

The decision-based Boundary Attack Brendel et al. (2017) was applied *via* Foolbox Rauber et al. (2017) using 50 steps, unless stated otherwise. To evaluate the success of this attack, we measure the following score :

$$S(M) = \text{median}_i \left( \frac{1}{d} \left\| \eta_M(x_{\text{original}}^i) \right\|_2^2 \right)$$

introduced in Brendel et al. (2017) where $\eta_M(x_{original}) = x_{original} - x_{adversarial} \in \mathbb{R}^d$ is the adversarial perturbation found by the attack. We measure this score using 1,000 randomly sampled images from the test set. The final reported score is the average $S(M)$ calculated over 5 repetitions. All codes relative to adversarial evaluation are supplemented with this submission.

## A.2 Robustness to random noise for models trained to classify CIFAR-10 regularized using $S^{\text{PIXEL}}$

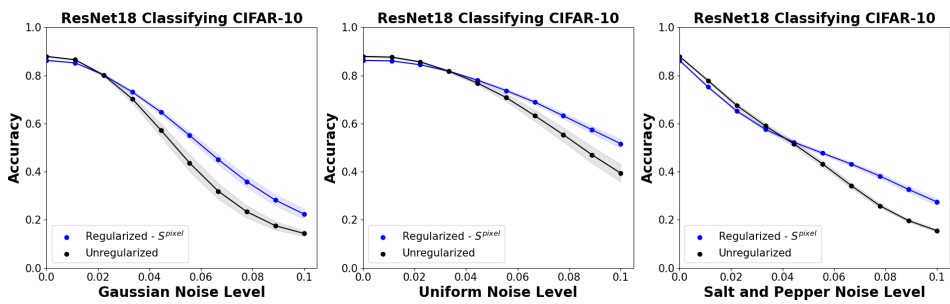

Figure 12: Robustness of a ResNet18 trained to classify grayscale CIFAR-10 and regularized on $S^{pixel}$ from grayscale images from ImageNet dataset to Gaussian, Uniform noise and Salt and Pepper noise. Error shades represent the SEM across seven seeds per model.

## A.3 Robustness to random noise for models trained to classify CIFAR-10 regularized using $S^{Th}$

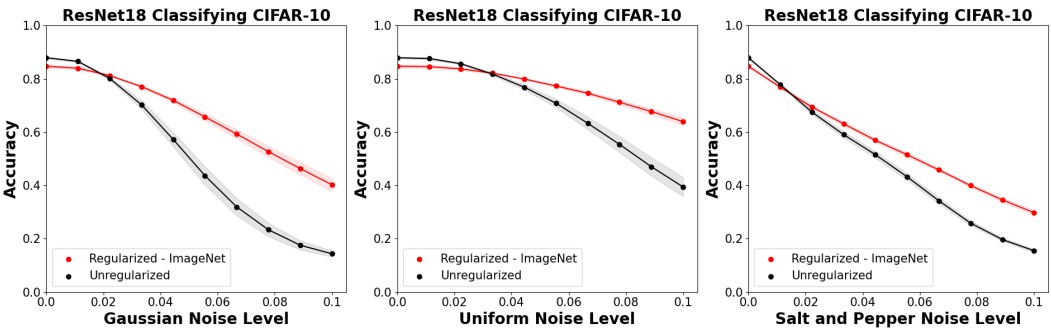

Figure 13: Robustness of a ResNet18 trained to classify grayscale CIFAR-10 and regularized on $S^{Th}$ from grayscale images from ImageNet dataset to Gaussian, Uniform noise and Salt and Pepper noise. Error shades represent the SEM across seven seeds per model.

## A.4 Robustness on image classification task for different classification-regularization datasets

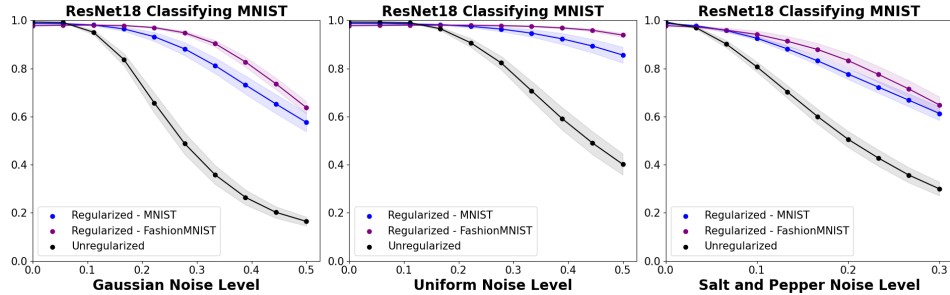

Figure 14: Robustness of a ResNet18 trained to classify MNIST and regularized on images from MNIST and FashionMNIST datasets to Gaussian noise, Uniform noise and Salt and Pepper noise. Error shades/bars represent the SEM across seven seeds per model.

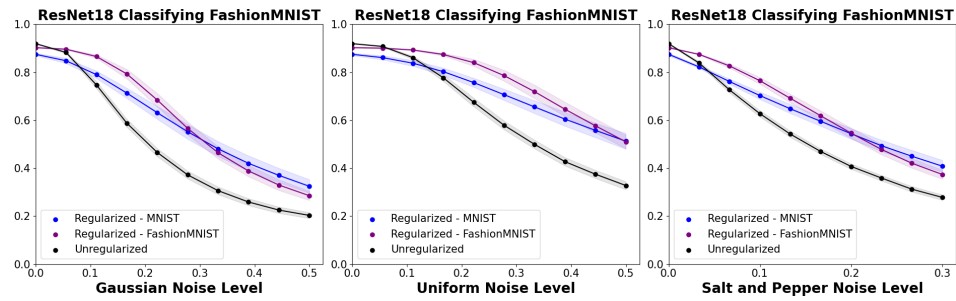

Figure 15: Robustness of a ResNet18 trained to classify FashionMNIST and regularized on images from MNIST and FashionMNIST datasets to Gaussian, Uniform and Salt and Pepper noise. Error shades/bars represent the SEM across seven seeds per model.

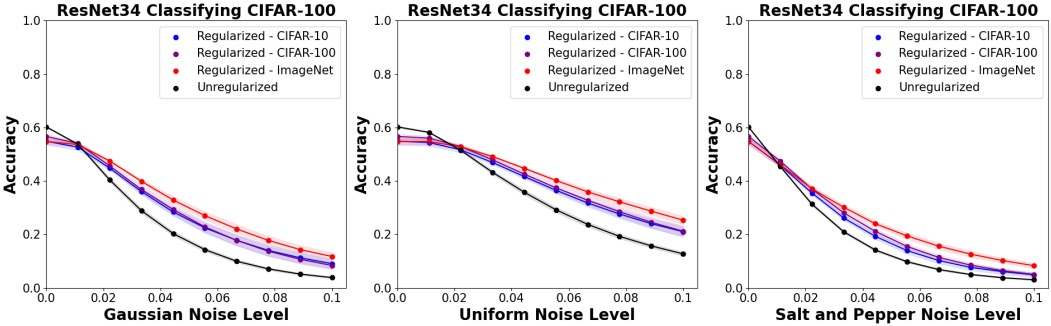

Figure 16: Robustess of a ResNet34 trained to classify grayscale CIFAR-100 and regularized on grayscale images from CIFAR-10, CIFAR-100, ImageNet datasets to Gaussian noise, Uniform noise and Salt and Pepper noise. Error shades/bars represent the SEM across seven seeds per model.

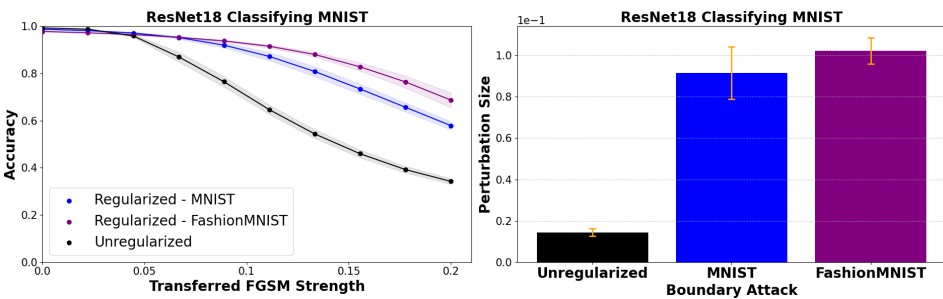

Figure 17: Robustess of a ResNet18 trained to classify MNIST and regularized on MNIST and FashionMNIST images to transferred FGSM Goodfellow et al. (2014) perturbations from an unregularized model, and a decision boundary attack Brendel et al. (2017). Error shades/bars represent the SEM across seven seeds per model.

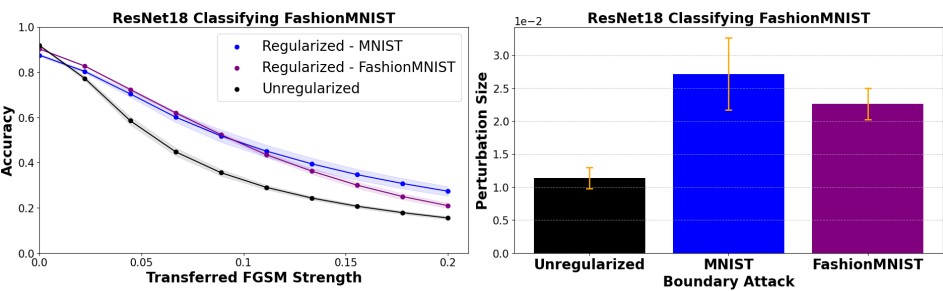

Figure 18: Robustess of a ResNet18 trained to classify FashionMNIST and regularized on images from MNIST and FashionMNIST to transferred FGSM Goodfellow et al. (2014) perturbations from an unregularized model, and a decision boundary attack Brendel et al. (2017). Error shades/bars represent the SEM across seven seeds per model.

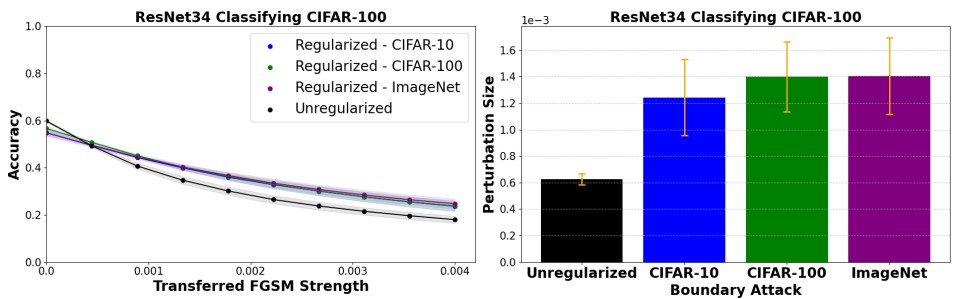

Figure 19: Robustess of a ResNet34 trained to classify grayscale CIFAR-100 and regularized on grayscale images from CIFAR-10, CIFAR-100, and ImageNet datasets to transferred FGSM Goodfellow et al. (2014) perturbations from an unregularized model, and a decision boundary attack Brendel et al. (2017). Error shades/bars represent the SEM across seven seeds per model.

## A.5 HYPERPARAMETERS USED FOR REGULARIZATION

The $\alpha, Th$ value pairs selected in our work yield an acceptable accuracy-robustness trade-off ($\frac{R_D}{R_0} > 1$ and $\frac{R_0}{U_0} \geq 0.9$) across attacks during regularization. These pairs are:

Table 1: Hyperparameters used for regularization

| Classification - Regularization | $\alpha$ | $Th$ |
|---|---|---|
| CIFAR-10 - CIFAR-10 | 10 | 0.8 |
| CIFAR-10 - CIFAR-100 | 10 | 0.8 |
| CIFAR-10 - ImageNet | 10 | 0.8 |
| CIFAR-100 - CIFAR-10 | 10 | 0.8 |
| CIFAR-100 - CIFAR-100 | 10 | 0.8 |
| CIFAR-100 - ImageNet | 10 | 0.8 |
| MNIST - MNIST | 4 | 0.2 |
| MNIST - FashionMNIST | 10 | 0.8 |
| FashionMNIST - MNIST | 10 | 0.4 |
| FashionMNIST - FashionMNIST | 10 | 0.8 |

We did not conduct an extensive hyperparameter sweep, so there are likely some $\alpha$ and $Th$ pairs that could yield stronger robustness.

## A.6 WEIGHTING CANDIDATE LAYERS

We here report the weights $\gamma_l$ obtained after training, for different models and dataset-combinations.

Table 2: Averaged weights for all candidate layers in a ResNet18 across seven seeds per model

| Classification - Regularization | $\gamma_1$ | $\gamma_5$ | $\gamma_9$ | $\gamma_{13}$ | $\gamma_{17}$ |
|---|---|---|---|---|---|
| CIFAR-10 - CIFAR-10 | $8.2 \times 10^{-3}$ | 0.56 | 0.38 | $5.2 \times 10^{-2}$ | 0 |
| CIFAR-10 - CIFAR-100 | 0.62 | 0.38 | 0 | 0 | 0 |
| CIFAR-10 - ImageNet | 0.62 | 0.38 | 0 | 0 | 0 |
| MNIST - MNIST | 0.5 | 0.125 | 0.25 | 0.125 | 0 |
| MNIST - FashionMNIST | 0.125 | 0.375 | 0.5 | 0 | 0 |
| FashionMNIST - MNIST | 0.75 | 0.125 | 0.125 | 0 | 0 |
| FashionMNIST - FashionMNIST | 0.143 | 0.286 | 0.571 | 0 | 0 |

Table 3: Averaged weights for all candidate layers in a ResNet34 across seven seeds per model

| Classification - Regularization | $\gamma_1$ | $\gamma_7$ | $\gamma_{15}$ | $\gamma_{27}$ | $\gamma_{33}$ |
|---|---|---|---|---|---|
| CIFAR-100 - CIFAR-10 | 0.375 | 0.375 | 0.25 | 0 | 0 |
| CIFAR-100 - CIFAR-100 | 0.875 | 0.125 | 0 | 0 | 0 |
| CIFAR-100 - ImageNet | 0.8 | 0.2 | 0 | 0 | 0 |

## A.7 RELEVANCE OF DIFFERENT SIMILARITY RANGES

In this subsection, we share an investigation regarding the target similarity ranges which matter the most for regularization. We define the following sets of regularization target pairs $i, j : S_-^{Th} := \{S_{ij} \mid S_{ij} < -Th \text{ or } |S_{ij}| < Th\}$, $S_+^{Th} := \{S_{ij} \mid S_{ij} > Th \text{ or } |S_{ij}| < Th\}$, $S_{\text{low}}^{Th} := \{S_{ij} \mid |S_{ij}| < Th\}$, $S_{\text{high}}^{Th} := \{S_{ij} \mid |S_{ij}| > Th\}$. We also define a target set depending on two thresholds : $Th_1 > Th_2 > 0$ such that $S_{\text{double}}^{Th_1, Th_2} := \{S_{ij} \mid |S_{ij}| < Th_2 \text{ or } |S_{ij}| > Th_1\}$.

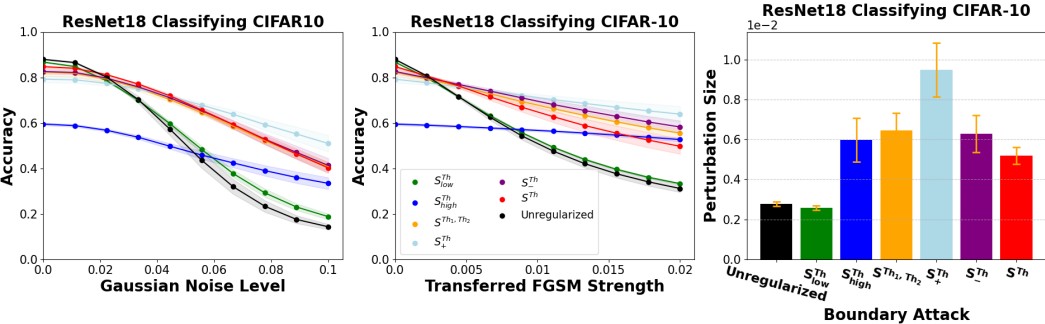

Figure 20: ResNet18 trained to classify grayscale CIFAR-10 and regularized with grayscale images from ImageNet for different regularization targets. Error shades/bars represent the SEM across seven seeds per model.

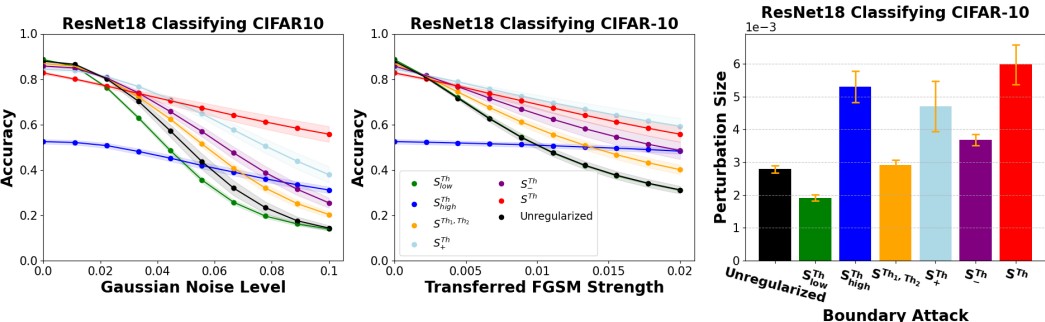

Figure 21: ResNet18 trained to classify grayscale CIFAR-10 and regularized with grayscale images from CIFAR-10 for different regularization targets. Error shades/bars represent the SEM across seven seeds per model.

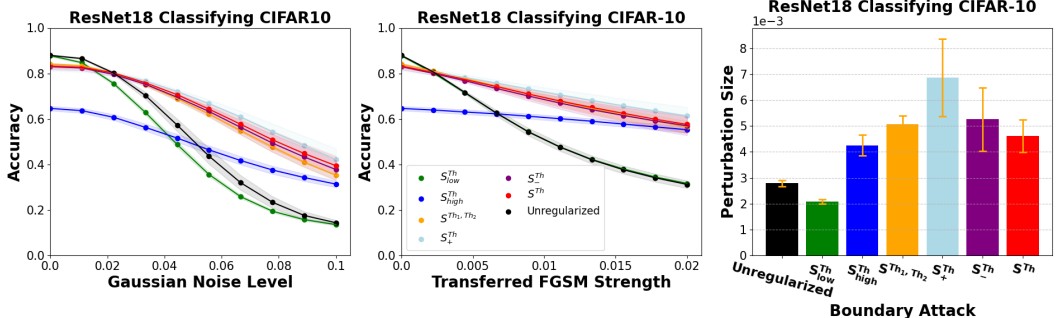

Figure 22: ResNet18 trained to classify grayscale CIFAR-10 and regularized with grayscale images from CIFAR-100 for different regularization targets. Error shades/bars represent the SEM across seven seeds per model.

## A.8 REGULARIZATION ON COLORED DATASETS

In Fig. 23 we show results using color CIFAR-100 as classification dataset, and color CIFAR-10, CIFAR-100 or ImageNet as regularization datasets. As seen, there is an increase in the model's robustness for all regularization datasets.

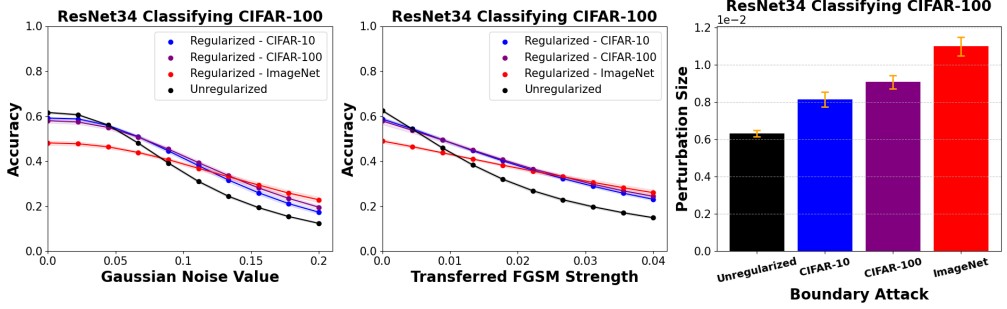

Figure 23: Robustness of a ResNet34 trained to classify colored CIFAR-100 regularized on colored images from different datasets : CIFAR-10 (blue), CIFAR-100 (purple) or ImageNet (red). For the decision-based Boundary Attack, we compute the median $L_2$ perturbation size, averaged across 1000 images, and 5 repeats. Error shades/bars represent the SEM across seven seeds per model.

### A.9 BENCHMARKING OUR METHOD AGAINST THOSE IN THE ROBUSTBENCH LEADERBOARD

We benchmarked a model trained to classify (colored) CIFAR-10 regularized on (colored) CIFAR-100 using our method, against 3 models referenced in the RobustBench Hendrycks & Dietterich (2019b) leaderboard - commonly used to systematically track the real progress in adversarial robustness - on CIFAR-10-C, at severity 5. We select $L_\infty$, $L_2$ and common corruption (thereafter CC) specific models denoted as Carmon2019Unlabeled Carmon et al. (2019), Engstrom2019Robustness Engstrom et al. (2019) , Modas2021PRIMEResNet18 Modas et al. (2022) respectively. We also benchmark against the 'standard' model available on RobustBench. As we see in Fig. 24, our regularized model performs better than the standard model reflecting the robustness gain, but does not reach the same performance as state of the art networks. We acknowledge that our method does not beat the state of the art methods in the adversarial robustness literature. The aim of our method is not to beat the state of the art methods, but rather to show that our method which is based on the neural regularizer in Li et al. (2019) can be equally effective without the need to use expensive neural data.

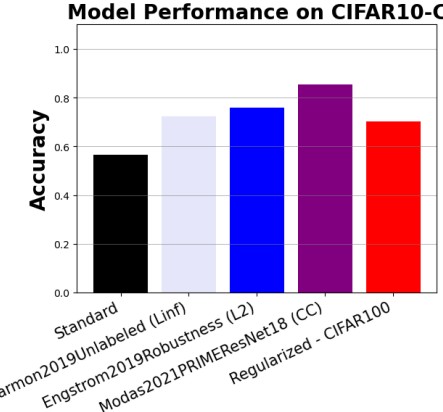

Figure 24: Benchmarking a model trained to classify (colored) CIFAR-10 regularized on (colored) CIFAR-100 using our method, against 3 models referenced in the RobustBench Hendrycks & Dietterich (2019b) leaderboard - commonly used to systematically track the real progress in adversarial robustness - on CIFAR-10-C, at severity 5. We select $L_\infty$, $L_2$ and common corruption (thereafter CC) specific models denoted as Carmon2019Unlabeled Carmon et al. (2019), Engstrom2019Robustness Engstrom et al. (2019) , Modas2021PRIMEResNet18 Modas et al. (2022) respectively. We selected our most robust model on CIFAR10-C for such benchmarking. We also benchmark against the 'standard' model available on RobustBench.

### A.10 FOURIER SPECTRA OF COMMON CORRUPTIONS

Following the approach of Li et al. (2023), we divided the common corruptions present in CIFAR-10-C into three categories based on their corresponding dominating frequencies whether they belong to low, medium or high frequency ranges. Low frequency corruptions are composed of 'snow', 'frost', 'fog', 'brightness', 'contrast' corruptions. Medium-frequency corruptions are composed of 'motion blur', 'zoom blur', 'defocus blur', 'glass blur', 'elastic transform', 'jpeg compression' and 'pixelate' corruptions. High-frequency corruptions are composed of 'gaussian noise', 'shot noise' and 'impulse noise' corruptions. For completeness, we reproduced the Fourier transformation of the common corruptions as shown below.

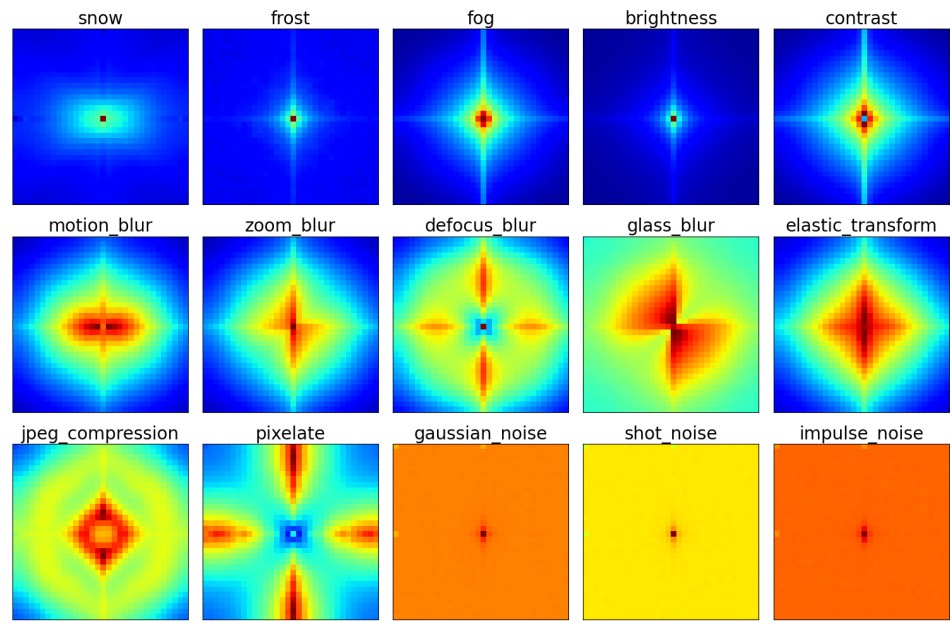

Figure 25: Fourier transform of the corruptions in CIFAR-10-C at severity 3, ordered by groups of frequency ranges from low, to medium to high. For each corruption, we compute $\mathbb{E}[|F(C(X) - X)|]$ by averaging over all test images, where $F$ denotes a discrete Fourier transformation, $X$ denotes an original image, and $C(X)$ its corrupted counterpart. We apply a logarithm $x \mapsto \log(1 + x)$ for visualization purposes.

