# OpenReview forum: "A Brain-Inspired Regularizer for Adversarial Robustness"
_ICLR.cc/2025/Conference — Submitted to ICLR 2025_

### Official Review · Reviewer_YD33 · 2024-11-01

**Soundness:** 3
**Presentation:** 2
**Contribution:** 1
**Rating:** 3
**Confidence:** 5

**Summary:**

The paper introduces a new, brain-inspired regularizer for CNNs that enhances robustness against adversarial attacks without needing neural data. By using pixel-based similarity measures instead of neural recordings, the proposed method simplifies implementation and makes the regularization process computationally efficient. Tests across several datasets and adversarial attacks (e.g., Gaussian noise, FGSM, and Boundary Attacks) show that this approach achieves significant robustness. Though it doesn’t outperform the latest adversarial defenses on all fronts, it holds promise for practical applications requiring efficient and accessible methods for adversarial robustness. The findings highlight how neuroscience-inspired regularizers can improve machine learning models without heavy data dependencies.

**Strengths:**

The strengths of this paper include an accessible regularizer that enhances robustness without neural data and computational efficiency by avoiding data augmentations.
I find the brain-inspired motivation of the paper also worthwhile.

**Weaknesses:**

There are many weaknesses.
The authors only tested on CIFAR. I could not see any tests comparing with the large amount of defenses available today.
The paper is heavily based on the paper from Li 2019.
I find the presentation to be lacking. There are no figures explaining the bioinspiration or the motivation behind the method,  or even how does this differs from the work from Li 2019.
The improvement from regularization looks very similar to the ones achieved by simple adversarial training.
The choice of attacks could. have been more diverse. L0 and Linfty attacks with both white and black box types are the standard for evaluation.

**Questions:**

How does the method presented compare with current defenses?

---

> ### Author Response · Authors · 2024-11-20
> **Responding to the Reviewer's comments.**
>
> We thank the reviewer for the comments.
>
> 1."The authors only tested on CIFAR."
>
> For the classification task, in addition to CIFAR-10, which is presented in the main text, we test on the following datasets: grayscale CIFAR-100 (Appendix A.4), color CIFAR-100 (Appendix A.8), MNIST (Appendix A.4) and FashionMNIST (Appendix A.4). This is mentioned on lines 246-249. For regularization, we used the same datasets and ImageNet as stated in the relevant figure captions.
>
> 2. "The paper is heavily based on the paper from Li 2019. I find the presentation to be lacking. There are no figures explaining the bioinspiration or the motivation behind the method, or even how does this differs from the work from Li 2019."
>
> Our paper significantly differs from the work of Li et al. (2019) in its goals, method and contributions. Li et al. (2019) used brain recordings directly to regularize artificial neural networks. This is very costly, because brain data is hard to collect, and not widely available. Here, we introduce a new regularizer, motivated by the neural recordings from the brain, that is computed using image pixel similarities, eliminating the need for neural recordings. Our goal is to extract the computational principle behind the regularization provided by brain recordings.
>
> The biological motivation behind our regularizer is presented in Fig. 2. In Fig. 2 we plot the similarities obtained from neural data and from image pixels. They are highly similar and correlated. The value of the correlation coefficient is r=0.83. This motivates us to use pixel similarities as a proxy for neural similarities.
>
> 3."The choice of attacks could have been more diverse. L0 and Linfty attacks with both white and black box types are the standard for evaluation."
>
> We thank the reviewer for this comment. Since our main goal is to demonstrate that our new regularizer captures the computational essence of regularization via brain recordings, we mainly use the adversarial robustness tests used in Li et al. (2019), where the authors used neural recordings from the brain for regularization. We show that all the results obtained by regularizing using explicit neural recording can be replicated using this new bio-inspired regularizer. In addition, we have also tested our method on SimBA attack, using grayscale datasets to allow us to compare our results with those obtained using the method in Li et al. (2019). For this attack we find that our regularization method provides a small enhancement in robustness compared to the unregularized model. But most importantly, we find that our results are similar to the results obtained from regularizing with brain neural recordings (Li et al., 2019) ). We will add a comment on this result in the paper, and add the figure to the supplementary material.

---

### Official Review · Reviewer_k7ZX · 2024-11-03

**Soundness:** 2
**Presentation:** 1
**Contribution:** 2
**Rating:** 3
**Confidence:** 3

**Summary:**

Inspired by the work of Li et al. (2019) and guided by the observation that pixel similarities in the original image exhibit a pattern similar to neural recordings, the authors propose a new regularization method to improve neural network robustness with a significant reduction in computational requirements. As claimed by the authors, the proposed method also applies to colour datasets and has been tested on ResNet18 using CIFAR10, CIFAR100, and ImageNet with Gaussian noise and some black-box attacks.

**Strengths:**

1. The idea presented in the paper is interesting.
2. Lots of experiments have been conducted to support the conclusions.
3. The paper considers various aspects of robustness, including both *adversarial robustness* and *common corruptions*.

**Weaknesses:**

1. There are lots of obvious grammar, abbreviation, and citation errors in the paper. Aside from the grammar mistakes, such as in line 12 of the Abstract where it says "... but they tend to sensitive ..." which should be "... they tend to be sensitive...", nearly all equation references are incorrectly formatted. It should be "Equation" or "Eq." instead of "eq.equation." Additionally, please check the references you've used. For instance, on lines 83-84 of page 2, you incorrectly cite Szegedy et al. (2013) ... Madry et al. (2017) without proper formatting. A parenthesis is likely needed here; please refer to the formatting instructions in section 4.1. You should review the paper at least once before submission.

2. The writing and organization of the paper are very vague and confusing. In the Introduction section on page 2, you list six contributions of your work. However, some of these contributions are too trivial to be included as a separate item. For example, in the fourth contribution, you state: 'We assess the robustness of the regularized models to common corruptions using the CIFAR-10-C dataset (Hendrycks & Dietterich, 2019a).' I do not believe this alone is substantial enough to be listed as a contribution. Some contributions could be combined. In the last contribution, you mention: 'We show that our regularization method primarily protects against high-frequency perturbations…,' which explains the mechanism behind your proposed method but does not stand alone as a contribution. Instead, the sentence on lines 74-75 serves more as a contribution but is not included in the contributions list.

3. Outdated related works. In the related works section, the most recent work you mention on adversarial attacks is from 2017, which was published seven years ago. You should include at least one recent study. Additionally, the most recent work across all of your related works is Kireev et al. (2022), which is only mentioned once. I suggest incorporating a broader scope of more up-to-date works.

4. Some concepts are not clearly explained. On page 3, lines 137-138, you mention S_{i,j}^{target} as the target's pairwise cosine similarity, but it is unclear what 'target' refers to.

5. Some of your claims need clearer evidence. On page 4, lines 162-166, you mention a similarity in the pattern of correlation between neural representational similarity and image pixel similarity, but the only evidence provided is Fig. 2, which is not convincing on its own. Additionally, your explanation of Fig. 2 is quite vague. For example, in the leftmost figure of Fig. 2, I see r=0.83, but there is no explanation provided for this value.

**Questions:**

1. As stated in your second contribution, 'We show that our regularizer drives the network to be more robust to a wide range of black-box attacks,' and since all your experiments are based on ResNet18, I am curious whether this method would work for different architectures, such as MLPs.

2. Since FGSM is $L_infty$ based and considered one of the weakest attack methods, I am curious whether your proposed method would still be effective against more advanced transfer attacks.

3. In Section 4.6, you attempt to demonstrate that your method is computationally inexpensive, but it is quite confusing to understand why this is the case from Fig. 8. Could you provide any plots related to computational complexity or time to support your argument?

---

> ### Author Response · Authors · 2024-11-20
> **Responding to the Reviewer's comments**
>
> We thank the reviewer for the comments.
>
> 1."There are lots of obvious grammar, abbreviation, and citation errors ...."
>
> Thank you for pointing these errors out. We apologize for them, and will fix them. We did review the paper multiple times, but those errors probably occured when editing the paper right before submission, they should not have occurred.
>
> 2. "The writing and organization of the paper are very vague and confusing ...."
>
> Thanks for this comment. You are right. We will take your suggestions into account and revise.
>
> 3. "Outdated related works. In the related works section, the most recent work you mention on adversarial attacks is from 2017, which was published seven years ago. You should include at least one recent study. Additionally, the most recent work across all of your related works is Kireev et al. (2022), which is only mentioned once. I suggest incorporating a broader scope of more up-to-date works."
>
> Thank you for pointing this out. We will update.
>
> 4. "Some concepts are not clearly explained. On page 3, lines 137-138, you mention $S_{i,j}^{target}$ as the target's pairwise cosine similarity, but it is unclear what 'target' refers to."
>
> We will pay extra attention to clarifying concepts in the next round of edits.
>
> With respect to your specific question about "target". We use the word "target" to represent the value of similarity that needs to be   achieved. We provide explanations for it on lines 149, 170 and 207. We will further clarify.
>
> 5. "Some of your claims need clearer evidence. On page 4, lines 162-166, you mention a similarity in the pattern of correlation between neural representational similarity and image pixel similarity, but the only evidence provided is Fig. 2, which is not convincing on its own. Additionally, your explanation of Fig. 2 is quite vague. For example, in the leftmost figure of Fig. 2, I see r=0.83, but there is no explanation provided for this value."
>
> Thanks for pointing these out. We will clarify further.
>
> First, r=0.83 refers to the Pearson Correlation Coefficient in Fig 2. We apologize for this omission.
>
> On lines 162-166, we strive to make the following points to promote our regularizer. First, we note that the visual system is hierarchical with increasingly more abstract representations as one goes up the hierarchy. V1 sits at the bottom of this hierarchy. Therefore, we expect similarities there to be closer to pixel level similarities. This observation motivates us to compare neural representational similarities and image pixel similarities. We observe a very high correlation in the standards of any neuroscience dataset (quantified by r = 0.83), which motivates us to introduce our own image-pixel-similarity-based regularizer.
>
> 6."As stated in your second contribution, 'We show that our regularizer drives the network to be more robust to a wide range of black-box attacks,' and since all your experiments are based on ResNet18, I am curious whether this method would work for different architectures, such as MLPs."
>
> Good suggestion! Although we stay within the CNN architecture, we show results using ResNet18, as well as ResNet34. Results for ResNet34 are presented in the supplementary results and we mention on lines 244-245 in the paper. We will consider more variation in architectures.
>
> 7. "Since FGSM Linifty is  based and considered one of the weakest attack methods, I am curious whether your proposed method would still be effective against more advanced transfer attacks.
>
> Thanks for this suggestion. Since our goal is to compare our regularization method to that of Li et al. (2019) which uses neural recordings, we mainly test on the attacks they used. In addition, we have also tested our method on SimBA attack, using grayscale datasets to allow us to compare our results with those obtained using the method in Li et al. (2019). For this attack we find that our regularization method provides a small enhancement in robustness compared to the unregularized model. But most importantly, our results are similar to the results obtained from regularizing with brain neural recordings (Li et al. , 2019). We will add a comment on this result in the paper, and add the figure to the supplementary material.
>
> 8."In Section 4.6, you attempt to demonstrate that your method is computationally inexpensive, but it is quite confusing to understand why this is the case from Fig. 8. Could you provide any plots related to computational complexity or time to support your argument?"
>
> In Fig.8 we show that the method works for small regularization batch size, and in Fig.9 we show that it works for small number of regularization images. In addition, it does not require special image augmentations. These factors reduce the computational resources and time needed to handle the data and train the network. Although we don't provide the quantities you mentioned, we believe the above evidence supports the statement that the method is computationally inexpensive.

---

### Official Review · Reviewer_oT1f · 2024-11-04

**Soundness:** 2
**Presentation:** 3
**Contribution:** 2
**Rating:** 3
**Confidence:** 3

**Summary:**

The paper inspects an existing neural regularizer to establish that the idea can be used for image pixel similarities without the use of actual neural recordings. Based on this observation, the authors propose a new regularization method based on two hyperparameters and eliminating the need for neural recordings. It is evaluated for different adversarial attacks and grayscale and colored datasets. Based on the selected threshold value, the target similarity matrix is calculated. The method is tested against transferable FGSM and Boundary attack as well as common image corruptions.

**Strengths:**

- The proposed method eliminates the need for neural recordings, which are generally expensive.
- Well-motivated, intuitive, and simple method to eliminate the actual neural recordings.
- Computationally light method.

**Weaknesses:**

- The authors mention, "our aim is not to come up with the best adversarial defence ..." on lines 519-521, which contradicts the title and the paper content that gives an impression the proposed method is to improve the adversarial robustness of the existing networks.
- The authors should discuss the preliminary details/definitions they use frequently from the baseline paper. For example, what are the classification and regularization datasets?
- While the improvement on the common corruptions is good, the evaluation of adversarial robustness is incomplete. The method is only tested against transferable FGSM and Boundary attacks. More robust evaluation with stronger white-box [1, 2] and black-box attacks [3] should be done to claim adversarial robustness.
- Evaluation is only limited to one CNN architecture - ResNet18. More evaluation of diverse architectures and models like WideResNet will provide additional evidence for the claims.
- In Section 2, the adversarial attacks mentioned are very old. The newer, more effective, and stronger attacks are not included. Similar observations are made for the adversarial defense subsection in Section 2.
- Minor typos - "imageNet" instead of "ImageNet" on line 389, every equation is referred as "eq. equation " instead of "Eq. " or "Equation ",

[1] Croce, Francesco, and Matthias Hein. "Reliable evaluation of adversarial robustness with an ensemble of diverse parameter-free attacks." International conference on machine learning. PMLR, 2020.
[2] Mądry, Aleksander, et al. "Towards deep learning models resistant to adversarial attacks." stat 1050.9 (2017).
[3] Andriushchenko, Maksym, et al. "Square attack: a query-efficient black-box adversarial attack via random search." European conference on computer vision. Cham: Springer International Publishing, 2020.

**Questions:**

My main concern is the contradictory claims in the paper. Please refer to the weakness section for more questions and details.

---

> ### Author Response · Authors · 2024-11-20
> **Responding to the Reviewer's comments**
>
> We thank the reviewer for the comments.
>
> 1. "The authors mention, "our aim is not to come up with the best adversarial defence ..." on lines 519-521, which contradicts the title and the paper content that gives an impression the proposed method is to improve the adversarial robustness of the existing networks."
>
> We thank the reviewer for this comment. It didn't occur to us that the title could be interpreted this way. We will consider changing it.
>
> With regards to the paper content, that we are doing a lot of adversarial tests might give the impression that our goal is to come up with a state-of-the-art adversarial defense. However, we are doing all these tests because we want to make a thorough comparison to the regularization method of Li et al. (2019). We will clarify this point.
>
> 2. "The authors should discuss the preliminary details/definitions they use frequently from the baseline paper. For example, what are the classification and regularization datasets?"
>
> Thank you for this suggestion. We will clarify.
>
> With regards to classification and regularization dataset, we discuss these terms on line 156 briefly, but we will expand. In short, the classification dataset, is the dataset we use for the classification task, and which contributes to $L_{task}$. The regularization dataset, is the dataset that we use to compute the similarity term in the regularizer $L_{sim}$. This distinction is made because we can use potentially separate datasets for classification and regularization. We show that our regularization method is flexible regarding the choice of the regularization dataset, and that one can use the same dataset for classification and regularization.
>
> Also, in the body of the paper, we mention in every experiment in figure titles and captions which dataset we use for the classification task (the classification dataset), and which dataset we use in the regularizer (the regularization dataset).
>
> 3."While the improvement on the common corruptions is good, the evaluation of adversarial robustness is incomplete. The method is only tested against transferable FGSM and Boundary attacks. More robust evaluation with stronger white-box [1, 2] and black-box attacks [3] should be done to claim adversarial robustness."
>
> We would like to stress again that the main goal of the paper is to show that our bio-inspired regularizer is a substitute for regularizing directly via brain recordings. We believe we have already shown substantial evidence for this.
>
> In addition, we have also tested our method on SimBA attack, using grayscale datasets to allow us to compare our results with those obtained using the method in Li et al. (2019). For this attack we find that our regularization method provides a small enhancement in robustness compared to the unregularized model. But most importantly, we find that our results are similar to the results obtained from the regularizing with brain neural recordings (Li et al., 2019). We will add a comment on this result in the paper, and add the figure to the supplementary material.
>
> 4. "Evaluation is only limited to one CNN architecture - ResNet18. More evaluation of diverse architectures and models like WideResNet will provide additional evidence for the claims."
>
> Thank you for the comment. Although we stay within the CNN architecture, we show results not only for ResNet18, we also show results for ResNet34 in the supplementary results, as we point out on lines 244-245 in the paper.

---

### Meta-Review · Area_Chair_oE5x · 2024-12-17

**Metareview:**

This paper proposes a brain-inspired regularizer that replaces neural recordings with image pixel similarities to enhance adversarial robustness in CNNs. Reviewers appreciated the novelty of removing the dependency on neural data and the potential computational efficiency of the approach. However, several concerns were raised: the evaluations were limited to weaker and outdated adversarial attacks (e.g., FGSM and Boundary attacks), with no testing on modern white-box or stronger black-box attacks. Experiments were restricted to ResNet18, with limited generalization to other architectures. Reviewers also flagged significant issues with writing, including grammar errors, vague explanations of concepts like "target" similarity, and poor formatting. Additionally, the related work section was seen as outdated, lacking references to recent adversarial defenses. While the idea is promising, the submission requires clearer presentation, more rigorous evaluation, and stronger positioning against current methods to meet the standards of ICLR 2025.

**Additional Comments On Reviewer Discussion:**

The reviewers share common concerns about the lack of enough evaluations and writing mistakes. The authors promise to add new results and revise the paper but these are not included in the rebuttal. No subsequential discussions were raised among the reviewers, though the AC requested so. It seems that their main concerns remain.

---

### Decision · Program_Chairs · 2025-01-22

Reject